# Protein Sequence Generation Model

## 1 Introduction

Protein sequence generation is a key area in computational biology, where the objective is to generate sequences with specific functional and structural features. Deep learning models have made substantial progress in this domain, but still face significant challenges in handling long sequences and modeling biological dynamics. While large language models have shown impressive results, their computational requirements often limit accessibility in academic research settings.

This study focuses on implementing and evaluating a small-scale Mamba architecture for protein sequence generation. The Mamba architecture offers several advantages through its selective state-space updates: linear computational complexity, efficient handling of long-range dependencies, and natural alignment with protein dynamics. Our work aims to demonstrate that competitive performance can be achieved with reduced computational resources, making advanced protein sequence modeling more accessible to the research community.

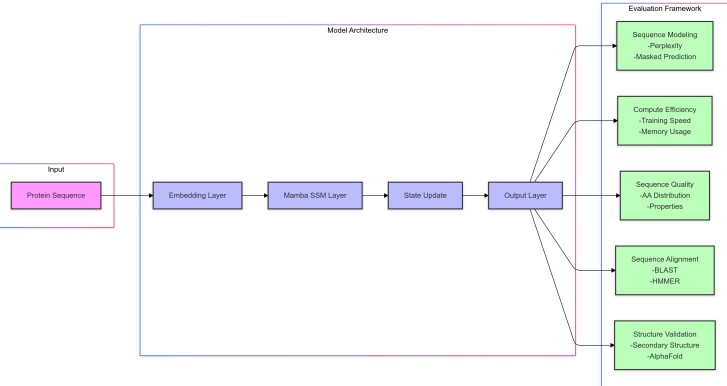

Figure 1: Overview of our proposed approach showing (A) Mamba architecture for protein sequence modeling, (B) multi-dimensional evaluation framework, and (C) experimental workflow.

## 2 Related Work

The field of protein language modeling has seen significant advances through various architectural approaches. Early works like UniRep [3] demonstrated that relatively small models using mLSTM architecture could effectively capture both evolutionary and biophysical properties. TAPE [1] provided comprehensive benchmarks showing that models with 10-20M parameters can achieve competitive results across multiple tasks, establishing a foundation for efficient protein modeling.

Structure-aware approaches emerged through works like Bepler and Berger [2], which incorporated structural information via multi-task learning and introduced the soft symmetric alignment mechanism.

Submitted to 38th Conference on Neural Information Processing Systems (NeurIPS 2024). Do not distribute.

The generative capacity of protein language models was significantly advanced by ProtGPT2 [5], which demonstrated successful unsupervised sequence generation, though at the cost of substantial computational resources.

Recent innovations in state space models have opened new possibilities for efficient protein modeling. ProtMamba [6] pioneered the application of the Mamba architecture to proteins, demonstrating that alignment-free approaches could effectively capture homology information. PTM-Mamba [7] extended this framework to handle post-translational modifications, highlighting the architecture's versatility. The core Mamba design [4] offers several key advantages for protein sequence modeling:

- Linear computational complexity O(L) versus transformers' quadratic O(L²)
- Efficient handling of long-range dependencies through selective state updates
- Reduced memory requirements enabling training on limited compute resources
- Natural alignment with protein dynamics through state-based modeling

Based on these developments, we focus on evaluating small-scale models (8-15M parameters) that balance performance with efficiency: UniRep-Small (mLSTM), TAPE-LSTM (bidirectional LSTM), Mini-Transformer, and our proposed Small Mamba model.

## 3  Proposed Model

Our implementation of the Mamba architecture leverages selective state-space modeling to efficiently process protein sequences. The core model is described by:

$$\dot{h}(t) = Ah(t) + Bu(t), \quad y(t) = Ch(t)$$

with a selective state update mechanism:

$$s_t = \sigma(W_s h_t + b_s) \odot s_{t-1} + (1 - \sigma(W_s h_t + b_s)) \odot \tanh(W_h h_t + b_h)$$

This mechanism enables adaptive retention of historical information while maintaining linear complexity. Key design considerations include optimized parameter count to balance expressivity and efficiency, selective state updates for capturing long-range protein dependencies, adaptable hidden state dimension based on sequence complexity, and efficient batch processing for training acceleration.

## 4  Evaluation Framework and Expected Results

Our comprehensive evaluation framework integrates multiple aspects of protein sequence analysis and generation. For sequence modeling capability, we assess the models' proficiency through perplexity measurements across varying sequence lengths and masked token prediction accuracy, with particular attention to long-range interaction prediction at known binding sites. The computational efficiency analysis examines both training dynamics and resource utilization, comparing theoretical predictions with empirical measurements.

Sequence quality assessment combines multiple complementary approaches. We employ BLAST and HMMER for similarity scoring and domain conservation analysis, supplemented by Clustal Omega for multiple sequence alignment. Structure validation integrates both secondary structure prediction via PSIPRED and tertiary structure prediction through AlphaFold, with quality metrics including TM-score and RMSD. The framework also evaluates the biological plausibility of generated sequences through amino acid distribution analysis and physicochemical property conservation.

Our experiments utilize the UniRef50 dataset, carefully selecting 10 million sequences to ensure comprehensive coverage while maintaining computational feasibility. Training conditions and hyperparameters are standardized across all models to enable fair comparison. The experimental phase spans eight weeks, with systematic evaluation of each metric.

Key expected contributions include first comprehensive evaluation of small-scale Mamba for protein generation, novel insights into efficient protein sequence modeling, and practical guidelines for protein engineering under computational constraints. These findings will serve as a valuable reference for the protein engineering community, particularly in resource-constrained academic settings. Our work aims to demonstrate that advanced protein sequence modeling can be achieved with modest computational resources while maintaining competitive performance.

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
