# OpenReview forum: "[Proposal-ML]Protein Sequence Generation Model"
_tsinghua.edu.cn/THU/2024/Fall/AML — THU 2024 Fall AML Submission_

### Official Review · ~Juncheng_Yu1 · 2024-11-07
**Innovative Application of Mamba Architecture for Efficient Protein Sequence Prediction**

**Rating:** 8
**Confidence:** 3

**Review:**

## Summary

This paper explores the application of the Mamba architecture to protein sequence prediction, representing an innovative interdisciplinary practice in computational biology. The study leverages Mamba’s selective state-space updates, which offer advantages in computational efficiency and handling long-range dependencies, making this approach both innovative and feasible. The paper presents a comprehensive framework, including model evaluation metrics and a detailed experimental design, demonstrating a thoughtful approach to achieving high-performance protein sequence modeling.

## Strengths

- **Innovative Exploration of Mamba Architecture**: The application of Mamba architecture in protein sequence prediction is creative and shows promise for advancing the field. The selective state-space updates provide an efficient means to model protein sequences, which could be highly valuable for resource-constrained settings.

- **Importance of the Research Problem**: Protein sequence prediction is a vital area of study with broad implications for biological research and practical applications. Addressing this challenge could aid in advancing drug discovery and understanding protein functions, underlining the relevance of this research.

- **Thorough Literature Review and Feasibility**: The paper provides a well-rounded review of related work, demonstrating the authors' strong understanding of the field. The chosen approach appears feasible, with a clear plan for implementation and evaluation.

## Weaknesses

- **Small and Unclear Figures**: The figures in the paper are too small and lack clarity, making it difficult to interpret the visual information provided. Larger, more detailed figures would significantly improve the readability and comprehension of the model architecture and results.

- **Writing Quality**: The writing could benefit from improved clarity and logical flow. More precise and sophisticated word choices would enhance the paper’s presentation.

## Score

- **Soundness**: 8/10
- **Contribution**: 8/10
- **Presentation**:7/10

---

### Official Review · ~Peidong_Zhang1 · 2024-11-08
**Strengths and limitations of proposal**

**Rating:** 9
**Confidence:** 4

**Review:**

This proposal presents a small-scale Mamba architecture for protein sequence generation, focusing on efficient protein modeling with reduced computational resources. The Mamba model is designed to handle long-range dependencies and protein dynamics while maintaining linear complexity, offering a promising solution for resource-constrained research environments. The proposed evaluation framework covers multiple aspects, including sequence quality, computational efficiency, and biological relevance, with comprehensive metrics for assessment. However, the proposal lacks detailed information on how the Mamba model will perform in comparison to other established small-scale models, such as UniRep or TAPE, under identical conditions. Additionally, the scalability of the method for larger, more complex sequences remains unclear, and more discussion on the limitations of the proposed model would be helpful.

---

### Official Review · ~Yunghwei_Lai1 · 2024-11-08

**Rating:** 10
**Confidence:** 4

**Review:**

The project proposes using a small-scale Mamba architecture, which is a compelling choice. Mamba's selective state-space updates and linear computational complexity make it well-suited for handling long protein sequences and capturing complex dynamics.

This project has a well-defined goal, focusing on an accessible yet effective architecture for protein sequence generation. Its emphasis on reducing computational requirements is highly relevant for academia. Clarifying methodological and evaluation aspects would enhance the project, but overall, it’s a strong proposal with potential for impactful results in the computational biology community.

---

### Official Review · ~Matteo_Jiahao_Chen1 · 2024-11-11
**Review of "Protein Sequence Generation Model"**

**Rating:** 9
**Confidence:** 4

**Review:**

This proposal introduces an approach to protein sequence generation using the  Mamba architecture. The authors aim to reduce computational complexity while maintaining competitive performance in protein sequence modeling.
## Strengths:
1. **Comprehensive Evaluation:**
   The proposal uses a robust evaluation framework .

2. **Practical Implications for Academic Research:**
   By focusing on small-scale models, the paper presents a pathway for achieving state-of-the-art results in resource-constrained environments.

## Weaknesses:

1. **Lack of Detailed Comparison with Existing Methods:**
A direct comparison with transformer models and other state-space models would strengthen the argument for Mamba's advantages.

2. **Unclear figures and writing"**:
The caption of the figure could be more detailed, improving the comprehension of the model architecture.

---

### Official Review · ~Hector_Rodriguez_Rodriguez1 · 2024-11-11
**Review of “Protein Sequence Generation Model”**

**Rating:** 9
**Confidence:** 4

**Review:**

The authors propose using a small Mamba model for protein sequence generation.

- The introduction highlights the challenges of protein sequence generation and explains the unique characteristics of the Mamba architecture that make it competitive with LLMs while requiring fewer computational resources. However, the clarity of Figure 1 could be improved by using larger typography, and the caption should be revised to accurately describe the figure.

- The related work section provides a good understanding of the state-of-the-art in protein sequence generation. This section could benefit from more consistent use of terms like “protein language modeling,” “protein sequence generation,” and “protein sequence modeling,” which would enhance clarity.

- The proposed architecture incorporates multiple aspects that ideally should be disclosed in more detail, particularly the meaning of each parameter. Additionally, it would be valuable to clarify which aspects of the model are novel compared to previously mentioned implementations like ProtMamba or PTM-Mamba.

- The evaluation framework and expected results are thorough and well planned, and the dataset selection is justified in detail.

Overall, the proposal is clear and well-written, providing a good understanding of the problem and the future work.

---

### Official Review · ~Ruilin_Hu2 · 2024-11-12
**Proposal of paper "Protein Sequence Generation Model"**

**Rating:** 10
**Confidence:** 4

**Review:**

This proposal effectively explores the implementation of a small-scale Mamba architecture for protein sequence generation, addressing computational efficiency, long-sequence handling, and biological plausibility. The primary strengths include (1) innovative use of selective state-space updates, yielding linear complexity, (2) comprehensive evaluation across structural and functional metrics, and (3) accessibility through reduced computational demands. However, potential weaknesses are (1) limited scalability compared to larger models and (2) reliance on specific datasets, which may affect generalization. Overall, the proposal promises meaningful contributions to resource-efficient protein modeling.

---

### Official Review · ~Grace_Xin-Yue_Yi1 · 2024-11-12

**Rating:** 10
**Confidence:** 3

**Review:**

The proposal discusses the importance of protein sequence generation and the challenges associated with current models in terms of computational resources and sequence length handling. It clearly outlines the motivation for using the Mamba architecture, emphasizing efficiency and accessibility. The proposal also includes a well-researched review of related work, from foundational models like UniRep and TAPE to advanced models like ProtGPT2 and ProtMamba. The evaluation section is detailed and thorough, outlining various aspects such as sequence modeling capability, computational efficiency, and sequence quality.

---

### Official Review · ~ChenJian1 · 2024-11-12
**Brief review**

**Rating:** 9
**Confidence:** 4

**Review:**

The proposal presents a small-scale Mamba architecture-based protein sequence generation model, aiming to efficiently handle long sequences and simulate protein dynamics through selective state-space updates. The study emphasizes achieving competitive performance while reducing computational resource requirements, making advanced protein sequence modeling more accessible to the academic research community. With a multi-dimensional evaluation framework and experimental workflow, the study is expected to provide valuable references for the protein engineering field, especially in computationally constrained academic settings.

### Strengths:

①The proposed Mamba architecture offers linear computational complexity, efficient handling of long-range sequence dependencies, and natural alignment with protein dynamics.
②The model design considers parameter optimization to balance expressiveness and efficiency, adaptable hidden state dimensions based on sequence complexity, and efficient batch processing for training acceleration.
③The comprehensive evaluation framework covers multiple aspects of protein sequence analysis and generation, including sequence modeling capability, computational efficiency analysis, and sequence quality assessment.
④The experimental design is rigorous, using the UniRef50 dataset to ensure comprehensive coverage while maintaining computational feasibility.
### Weaknesses:

①Although the advantages of the Mamba architecture are proposed, the proposal lacks specific details on how to handle complex biochemical characteristics within protein sequences.
②While the experimental part is well-designed, there is a lack of assessment of the model's generalization ability, especially its performance on different types of protein sequences.
③The proposal is not clear enough on the model's scalability and the direction of future work.

---

### Official Review · ~Yuji_Wang4 · 2024-11-12
**Review of “Protein Sequence Generation Model”**

**Rating:** 9
**Confidence:** 3

**Review:**

The authors propose to explore applying small-scale Mamba models for protein sequence generation, which is expected to be an efficient approach than previous methods based on other architecture like transformers.

### Strengths
1. Topic selection: Both Mamba and protein sequence generation are popular research topics with high practical value, making the project a meaningful exploration.
2. Clear proposal structure：The proposal clearly defines the problem and provides a thorough review of related works. Meanwhile, the offered training and evaluation methods build a strong foundation for the project.

### Weaknesses
1. Lack of clarity: More detail is needed. For example, readers unfamiliar with the topics may not know why Mamba models are well-suited for this task and how they outperform transformers.
2. The figure may be confusing as readers can not easily understand the three parts mentioned in the caption.

---

### Official Review · ~Kaiwei_Zhang3 · 2024-11-12
**The format needs to be revised**

**Rating:** 8
**Confidence:** 3

**Review:**

**1. Summary:**

This research proposal presents a small-scale implementation of the Mamba architecture for protein sequence generation, targeting efficiency and accuracy in handling long protein sequences. The work aims to balance performance and computational efficiency, making advanced protein modeling techniques accessible for researchers with limited resources.



**2. Clarity:**

The proposal is mostly clear, yet some sections could benefit from more explanation, such as the formulas in *Proposed Method* part.



**3. Originality:**

The work is original in its focus on adapting the Mamba architecture for protein sequence generation at a smaller scale, making high-performance protein modeling more accessible.



**4. Significance:**

The proposal focuses on a crucial problem. If successful, it could significantly benefit researchers in resource-constrained settings, enabling them to achieve competitive results without extensive computational resources.



**5. Pros:**

* **Efficient model design.** The Mamba architecture’s linear complexity and adaptive state updates are promising for reducing computational requirements.
* **Practical application**. Targeting computational efficiency makes the approach accessible to a wider range of researchers, addressing a common limitation in the field.



**6. Cons:**

* **Formatting issue.** Names of the authors are not written. Text in figure 1 is too small. Index of each line should not be shown. A special font should be used for the big-O notations.

* **Lack of clarity.** The formulas in *Proposed Model* Section needs to be furthur illustrated. The meaning of the equations and symbols are not clear.

---

### Official Review · ~liyingxin1 · 2024-11-12
**Should show more difference between existed methods**

**Rating:** 8
**Confidence:** 4

**Review:**

There are some existed way to generate protein sequence. So maybe it is recommended to clearly articulate the specific objectives and expected outcomes of the study in the introduction to help readers better understand the significance and innovation of the research.

In the methodology section, more technical details could be provided regarding the specific implementation of the Mamba architecture and the selective state update mechanism to help readers better understand the implementation process and technical advantages.

It is recommended to include an experimental section to demonstrate the effectiveness and advantages of the proposed methods in practical applications, particularly in handling long sequences and modeling biological dynamics.

---

### Official Review · ~Zhu_Zhang6 · 2024-11-12
**Good proposal, well explained**

**Rating:** 9
**Confidence:** 3

**Review:**

**Summary:**

This proposal presents a protein sequence generation model based on the Mamba architecture, aiming to improve efficiency and accessibility for protein modeling. By leveraging the Mamba model's selective state-space updates, which offer linear computational complexity and efficient handling of long-range dependencies, the authors seek to generate high-quality protein sequences with reduced computational resources. They plan to evaluate the model’s performance across multiple dimensions, including sequence generation quality, structural accuracy, and computational efficiency, with a focus on making protein modeling more accessible to researchers with limited resources.

**Strengths:**
1. **Efficient Model Choice:** The use of Mamba architecture, known for its computational efficiency, addresses accessibility issues and makes complex protein modeling more feasible in resource-limited settings.
2. **Comprehensive Evaluation Framework:** The proposal outlines a multi-faceted evaluation framework, covering sequence quality, structural validation, and biological plausibility, which provides a thorough assessment of the model's capabilities.

**Weaknesses:**
1. **Evaluation of Long-Term Stability Missing:** There is no mention of testing the long-term stability or usability of generated sequences in practical protein engineering applications.

**Questions:**
1. How will the authors ensure that the generated sequences maintain functionality and stability in real biological contexts?
2. Is there a plan to compare the model’s performance with existing transformer-based protein models, specifically in terms of computational cost and sequence quality?

---

### Official Review · ~Fei_Long3 · 2024-11-12
**A Good Proposal But Requires Optimization on Layout**

**Rating:** 8
**Confidence:** 4

**Review:**

**Strengths**:

1. **Innovative Architecture Application:** The proposal demonstrates a significant strength by introducing the Mamba architecture to the field of protein sequence generation, which is a novel approach that could potentially revolutionize how computational biology handles protein modeling, especially given the architecture's ability to handle long sequences and model biological dynamics efficiently.

2. **Addressing Computational Accessibility:** The focus on reducing computational requirements while maintaining competitive performance is a strong point. This work could make advanced protein sequence modeling more accessible to a broader research community.

**Weakness**:

**Figure and Layout Optimization:** While the content of the proposal is strong, the document could benefit from improved figure and layout optimization for enhancing the clarity of these visual elements, which would significantly improve the proposal's impact.